# Two-year efficacy of varenicline tartrate and counselling for inpatient smoking cessation (STOP study): A randomized controlled clinical trial

Kristin V. Carson-Chahhoud[1,2]*, Brian J. Smith[2], Matthew J. Peters[3], Malcolm P. Brinn[1], Faisal Ameer[4], Kuljit Singh[2,5], Robert Fitridge[6], Simon A. Koblar[7], Jim Jannes[7,8], Antony J. Veale[9], Sharon Goldsworthy[10], Khin Hnin[11], Adrian J. Esterman[12]

1 Australian Centre for Precision Health, School of Health Sciences, The University of South Australia, Adelaide, South Australia, Australia, 2 School of Medicine, The University of Adelaide, Adelaide, South Australia, Australia, 3 Thoracic Medicine, Concord Repatriation General Hospital, Sydney, New South Wales, Australia, 4 Respiratory Medicine, Ipswich Hospital, Ipswich, Queensland, Australia, 5 Cardiology Department, Gold Coast University Hospital, Gold Coast, Queensland, Australia, 6 Division of Surgery, The Queen Elizabeth Hospital, Adelaide, South Australia, Australia, 7 Stroke Research Programme, University of Adelaide, Adelaide, South Australia, Australia, 8 Stroke Unit, The Queen Elizabeth Hospital, Adelaide, South Australia, Australia, 9 Respiratory Medicine Department, The Queen Elizabeth Hospital, Adelaide, South Australia, Australia, 10 Pharmacy Department, The Queen Elizabeth Hospital, Adelaide, South Australia, Australia, 11 Department of Respiratory and Sleep Medicine, Flinders Medical Centre, Adelaide, South Australia, Australia, 12 School of Nursing and Midwifery, The University of South Australia, Adelaide, South Australia, Australia

* Kristin.Carson-Chahhoud@unisa.edu.au

**Data Availability Statement:** Data cannot be shared publicly because of participant confidentiality reasons. Data are available from the

## Abstract

### Introduction

Varenicline tartrate is superior for smoking cessation to other tobacco cessation therapies by 52 weeks, in the outpatient setting. We aimed to evaluate the long-term (104 week) efficacy following a standard course of inpatient-initiated varenicline tartrate plus Quitline-counselling compared to Quitline-counselling alone.

### Methods

Adult patients (n = 392, 20–75 years) admitted with a smoking-related illnesses to one of three hospitals, were randomised to receive either 12-weeks of varenicline tartrate (titrated from 0.5mg daily to 1mg twice-daily) plus Quitline-counselling, (n = 196) or Quitline-counselling alone, (n = 196), with continuous abstinence from smoking assessed at 104 weeks.

### Results

A total of 1959 potential participants were screened for eligibility between August 2008 and December 2011. The proportion of participants who remained continuously abstinent (intention-to-treat) at 104 weeks were significantly greater in the varenicline tartrate plus counselling arm (29.2% n = 56) compared to counselling alone (18.8% n = 36; p = 0.02; odds ratio 1.78; 95%CI 1.10 to 2.86, p = 0.02). Twenty-two deaths occurred during the 104 week study

Royal Adelaide Hospital Human Research Ethics Committee and The Queen Elizabeth Hospital and Lyell McEwin Hospital Human Research Ethics Committee (contact via Health. CALHNResearchGovernance@sa.gov.au or Health. HumanResearchEthicsCommittee@sa.gov.au) for researchers who meet the criteria for access to confidential data.

**Funding:** The authors received no specific funding for this work.

**Competing interests:** The authors have read the journal's policy and have the following potential competing interests: KVCC was paid an honorarium and provided with economy airfares and accommodation by Pfizer Australia to present at the 2019 Smoking Exchange Summit in New South Wales where she spoke about 'cultural-specific issues in smoking cessation' and as an invited panellist in a plenary session about 'a national approach to smoking cessation'. In 2017 she received an honorarium and provided with economy airfares and accommodation to speak about 12-month results of the STOP trial at the annual Pfizer Australia conference in New South Wales, Hunter Valley. This does not alter our adherence to PLOS ONE policies on sharing data and materials. There are no patents, products in development or marketed products associated with this research to declare.

(n = 10 for varenicline tartrate plus counselling and n = 12 for Quitline-counselling alone). All of these participants had known or developed underlying co-morbidities.

## Conclusions

This is the first study to examine the efficacy and safety of varenicline tartrate over 104 weeks within any setting. Varenicline tartrate plus Quitline-counselling was found to be an effective opportunistic treatment when initiated for inpatient smokers who had been admitted with tobacco-related disease.

## Introduction

For each death caused by cigarette smoking, 30 people remain living with serious tobacco-related illnesses [1]. On a global scale disease burden is set to increase, with approximately 6 million deaths per year growing to predictions of 8 million by 2030 [2]. Varenicline tartrate acts on the α4β2 nicotinic receptor, being the same receptor targeted by nicotine inhaled from smoke [3]. Unlike nicotine replacement therapy, varenicline has a dual action by simultaneously easing cravings whilst blunting smoking associated reward and pleasure through partial nicotinic acetylcholine receptor agonist activity. It has a longer half-life of effect compared to nicotine replacement therapy, which may be important in the inpatient hospitalised setting [4]. Indeed, targeting inpatients during a period of hospital confinement offers an opportunistic environment to initiate smoking cessation interventions, as it provides an opportunity for patients to reflect on the progression of events resulting in hospitalisation, a bedside phone to ensure initial contact with Quitline counselling and an observation period for medication related adverse events.

In 2009 the United States Food and Drug Administration (FDA) released a black box warning for varenicline following reports of psychiatric side effects including hostility, agitation, depressed mood and suicidal ideation [5]. However, in 2016 a study by Anthenelli *et.al*, (Evaluating Adverse Events in Global Smoking Cessation Study: EAGLES), evaluated safety and efficacy of smoking cessation pharmacotherapy among 8,000 smokers with and without history of psychiatric disorders, identifying no association between varenicline and an increased incidence of clinically significant neuropsychiatric adverse events [6]. Subsequently, the joint FDA advisory committee voted to remove the box warning pertaining to neuropsychiatric side effects [7]. The FDA has also raised concerns about a possible increased risk of cardiovascular adverse events including cardiovascular-related death, nonfatal heart attack and non-fatal stroke [8]. Another recent study by Eisenberg *et.al*, (Evaluation of Varenicline in Smoking Cessation for Patients Post-Acute Coronary Syndrome: EVITA), conducted in 302 patients hospitalised with acute coronary syndrome identified similar adverse event rates within 30 days of study drug discontinuation between groups [9]. These studies supports our own 52 week findings demonstrating that varenicline is effective [10], well-tolerated and can be safely administered [11] among smokers admitted to hospital with acute tobacco related illnesses. However, none of these studies examine long-term (greater than 52 week) efficacy.

Only one 1971 German placebo-controlled trial evaluated the long-term efficacy (104 week follow-up) of cytisine with statistically significant benefits (odds ratio 1.77; 95%CI 1.29 to 2.43). No other study to date has evaluated any nicotinic acetylcholine receptor agonist beyond 52 weeks. Therefore, one of the secondary objectives of this study (and main objective of this manuscript) is to report efficacy and mortality of varenicline plus counselling compared to

counselling alone at 104 week follow-up for inpatients admitted to hospital following an acute smoking-related illness.

## Methods

### Trial design

This study was an open-label randomised, multicentre controlled clinical trial, with a 12 week treatment phase. A pre-specified protocol was published online (available via clinicaltrial.gov identifier NCT01141855)), complying with the ethical principles of the Declaration of Helsinki and approved by the respective hospital ethics committees (Royal Adelaide Hospital Human Research Ethics Committee: 080520 and The Queen Elizabeth Hospital and Lyell McEwin Hospital Human Research Ethics Committee: 2008012). All participants provided written informed consent prior to commencement of pharmacotherapy or data collection. All authors were involved in some aspect of study design and/or conduct and each author contributed to writing the manuscript.

### Study population

As per our previous publications [10, 11], participants were recruited between August 2008 and December 2011 from three tertiary hospitals being The Queen Elizabeth Hospital, Lyell McEwin Hospital and Royal Adelaide Hospital in South Australia. Patients presenting to hospital under disciplines of respiratory, cardiology, neurology and vascular medicine following a serious tobacco-related illness (as defined by the Centre of Disease Control [12]), were considered for recruitment. Participants were considered for inclusion if they were aged between 18 and 75 years, smoked at least 10 cigarettes on average per day over the preceding 12 months, had a plan of discharge to go home and had no contraindications to varenicline. Participants were excluded if they had cancer within the past seven years, renal impairment with creatinine clearance <30ml/min, had acute or pre-existing psychiatric illnesses including depression uncontrolled with medication, past-history psychosis or suicidal ideation, were pregnant or breastfeeding, were using other forms of nicotine replacement therapy or had used varenicline in the past 12 months. Patients with psychiatric illnesses who were stable on medication were considered for inclusion.

### Randomization, allocation concealment and blinding

Following identification of participants through health professional notification of a participant meeting the inclusion/exclusion criteria and an initial screen of patient medical records they were approached for participation whilst an inpatient. Following an opportunity to consider enrolment, patients signed the consent and completed the baseline questionnaire prior to randomization. A computer-generated simple randomization sequence generation with permuted blocks of 20 was used to assign participants in a 1:1 ratio to either 12 weeks of varenicline tartrate plus Quitline counselling or Quitline counselling alone. Allocation concealment occurred with the use of consecutively numbered opaque, sealed envelopes that were opened by study investigators following completion of all baseline data collection. Randomization and allocation concealment were performed by respiratory staff independent of the study. Participants and investigators were not blinded to treatment assignment.

### Interventions

All participants were provided with a quit assistance resource pack provided by the Cancer Council of South Australia including a booklet on smoking cessation entitled "Quit because

you can", smoke free stickers and fridge magnets listing tips to help manage with cravings. Initial Quitline contact was instigated by the project officer who recruited the participant at the patient's bedside directly following recruitment and randomization, or a Quitline counsellor called the patient at a time specified by the participant. The counselling programme employed the 5A approach (Ask, Assess, Advise, Assist and Arrange), consisting of eight scheduled call backs over a 12 week period of approximately 5–10 minutes duration. Participants randomised to the intervention group received the same Quitline counselling and resource pack in addition to varenicline tartrate, administered orally at 0.5mg per day for the first three days, 0.5mg twice daily for 4 days, then 1mg twice daily thereafter for a total treatment duration of 12 weeks. Participants were permitted up to 14 days following commencement of varenicline to set their target quit date. Further details in relation to the intervention and comparator arm have been described elsewhere[10, 11].

## Outcomes and data collection

The outcome of interest for this manuscript was continuous abstinence between weeks 2 and 104 (two year follow-up) with the overarching primary outcome for the trial of continuous abstinence between weeks 2 and 52 (reported elsewhere [10]). Continuous abstinence was defined as smoking ≤ five cigarettes in total by the follow-up period at 104 week (in line with recommendations for smoking cessation trials [13, 14]). Abstinence was by self-report with bio-chemical validation in a random sub-set of participants via exhaled carbon monoxide levels of ≤ 10 ppm. Secondary outcomes included adverse events during the 12-week treatment period compared to outpatient studies and all-cause mortality by 52 and 104 weeks. Other secondary outcomes related to hospital bed utilisation and health care costs, 7-day point prevalence and inpatient craving levels are reported elsewhere (cost effectiveness manuscript in production and [11]).

Baseline data collection occurred by the project officer recruiting the participant, prior to randomization, with follow-up data collection over the phone by a different project officer/investigator than the one who recruited the participant. Data was stored electronically in a password-protected database case report forms were stored in hard copy within a lockable filing cabinet. This was an open-label study with participants assigned to the intervention arm paying the full Pharmaceutical Benefits Scheme subsidised costs or concession costs of varenicline if assigned to that treatment arm.

## Statistical analyses

Sample size of 196 per arm was calculated to produce a 15% difference (45% vs. 30%) at 52 weeks, using a two group uncorrected chi-squared test with a 0.05 two-sided significance level, based on available literature [15–17]. This provided 80% power to detect the difference between the two arms, with additional adjustments for attrition and clustering (20%) with a rho of 0.02 [18]. Treatment efficacy at 104 week follow-up was not factored into statistical power calculation, as it was a secondary objective. Efficacy was defined as continuous smoking abstinence (less than five cigarettes) between week two and 104 week follow-up, calculated using a two-sided chi-squared test and Mann-Whitney U-test. Adjustments were made for differences in baseline data between medical disciplines (i.e., imbalance in people allocated to VT +C group in the vascular discipline). Analyses were based on intention to treat using statistical packages STATA version 11 and SPSS version 19. Participants lost to follow-up, withdrawn from the study or deceased during the study period were assumed to be smoking for the purpose of 104 week efficacy, regardless of smoking status at last contact period. Missing data

from participant questionnaires were excluded from analyses. Data presented as mean and standard deviation (SD) unless otherwise specified.

## Results

A total of 1959 patients were screened for eligibility between August 2008 and December 2011. Of the 392 participants recruited (n = 196 per arm), over 50% retention was achieved by 104 week follow-up (Fig 1). Baseline demographics were similar between groups (Table 1), with the exception of more participants randomized to varenicline tartrate plus counselling (VT +C) in the vascular discipline (n = 19) compared to counselling alone (C-alone; n = 7).

For the primary outcome of self-reported continuous smoking abstinence between weeks 2 and 104 (intention-to-treat), a statistically and clinically significant benefit in favour of the VT +C arm was observed (VT+C 29.2% n = 56 compared to counselling alone 18.8% n = 36; odds ratio 1.78; 95%CI 1.10 to 2.86; p = 0.02). Significance in favour of VT+C over C-alone for continuous abstinence was maintained at each follow-up period from four weeks, which became more significant following adjustment for baseline differences within disciplines (Fig 2), being

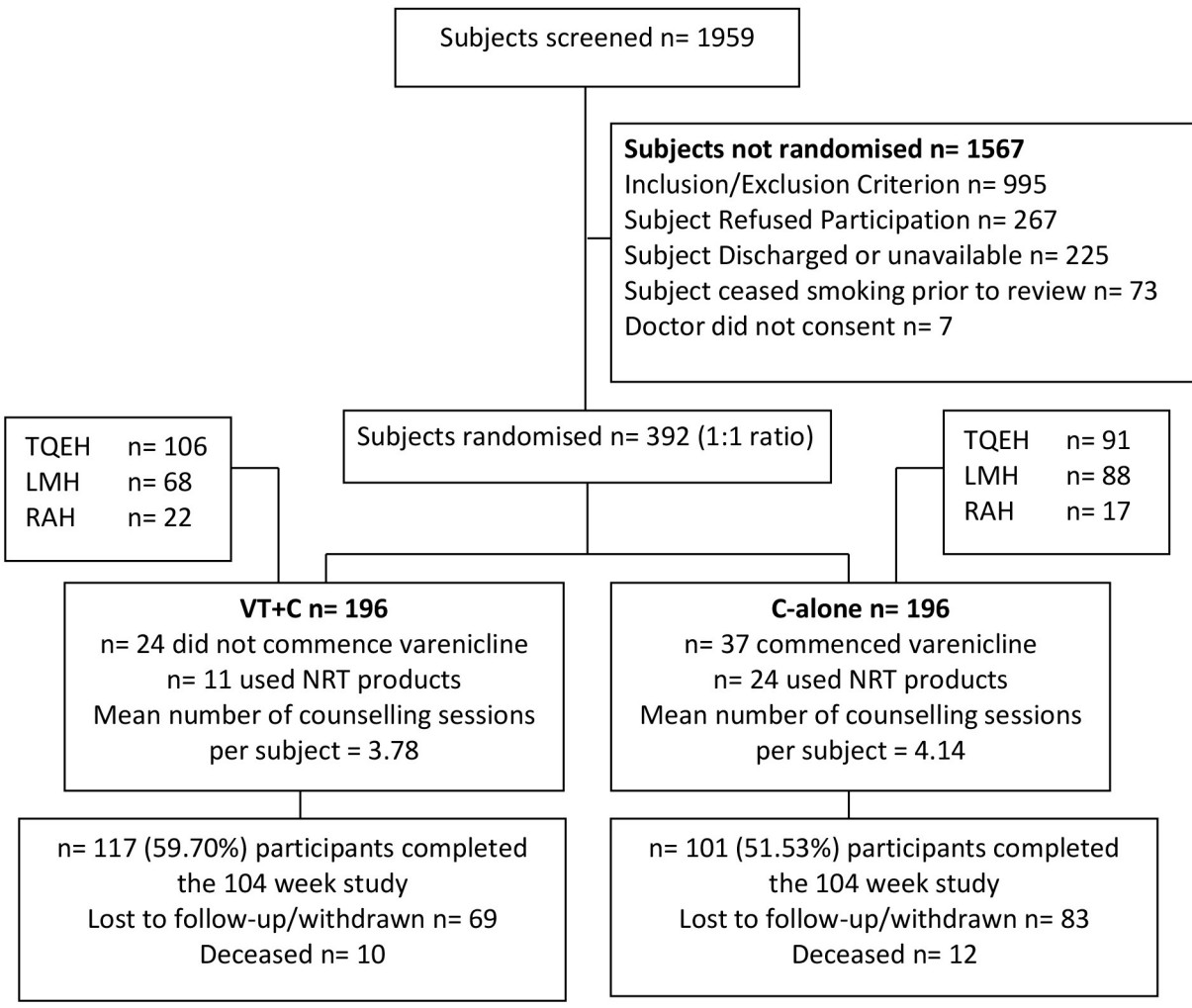

**Fig 1. Participant disposition (Consort flow diagram).**

**Table 1. Baseline demographic characteristics and smoking history.**

| Parameter | VT+C<br>n = 196 | C-alone<br>n = 196 |
|---|---|---|
| Age in years (range: 22–75) | 52.8 (2.89) | 53.7 (2.77) |
| Gender n(%) | | |
| Male | 138 (70.4) | 128 (65.3) |
| Female | 58 (29.59) | 68 (34.7) |
| Ethnicity n(%) | | |
| Caucasian | 186 (94.9) | 191 (97.5) |
| Aboriginal | 9 (4.6) | 4 (2.0) |
| Asian | 1 (0.5) | 1 (0.5) |
| Charleston Co-morbidity Index | | |
| Age unadjusted | 1.67 (0.94) | 1.57 (0.94) |
| Age adjusted | 2.64 (1.15) | 2.54 (1.17) |
| Pack years | 44.8 (3.93) | 45.7 (4.24) |
| Duration of smoking in years | 37.1 (2.93) | 37.7 (2.89) |
| Number of cigarettes per day in the past year | 24.9 (2.67) | 24.7 (2.89) |
| Fagerström score | 5.8 (1.3) | 5.4 (1.3) |
| Disciplines n(%) | | |
| Cardiology | 98 (50) | 97 (49.5) |
| Respiratory | 57 (29.1) | 65 (33.2) |
| Neurology | 22 (11.2) | 27 (13.8) |
| Vascular | 19 (9.7) | 7 (3.6) |

Mean and (standard deviation) unless otherwise specified; as per [11]

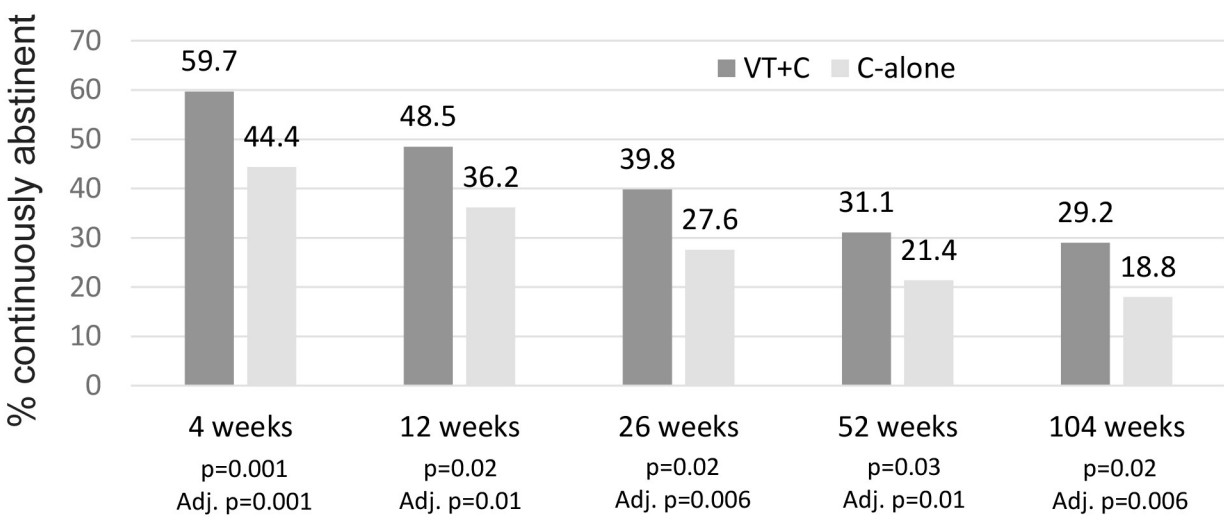

**Fig 2. Continuous smoking abstinence between weeks 4 and 104; Self-reported continuous abstinence defined as smoking $\leq$ 5 cigarettes between from week 2 to respective follow-up period.** Intention-to-treat analysis used (n = 196 per arm) with data above bar representing % abstinence. P-values are unadjusted and adjusted for baseline differences observed between disciplines.

an imbalance in allocation to the VT+C arm for people in the vascular discipline. For continuing smokers the average number of cigarettes smoked per day reduced from baseline to 104 week follow-up in both arms (VT+C 24.9 SD2.67 to 17.1 SD11.72 and counselling alone 24.7 SD2.89 to 15.4 SD8.82 respectively).

The most common adverse event reported by participants during the 12-week treatment phase was nausea with 16.3% in the VT+C group compared with 1.5% in C- alone. This was less than the prevalence of nausea reported by Pfizer (27%) among outpatient studies with healthy volunteers. Indeed, all adverse events reported among STOP trial participants was less than those observed by Pfizer reports (Table 2). The one exception was mortality. However, the STOP population all presented to hospital with acute illnesses and substantial co-morbidities. At both 52 and 104 week follow-up all-cause mortality was observed to be similar between groups (Table 3; not statistically significant).

## Discussion

This study showed superior treatment efficacy of varenicline plus counselling compared to counselling alone for self-reported continuous smoking abstinence at 104 week follow-up. Adverse events during the treatment period were lower in the STOP trial among the acute illness setting than side effects reported among outpatient 'healthy' volunteers. Mortality within 52 and 104 weeks were also reported to be similar between groups.

The STOP trial is the first study world-wide to examine the efficacy and safety of varenicline tartrate over 104 weeks within any setting as well as being the first study to examine administration of varenicline within the inpatient setting among acute smokers with tobacco related illnesses. It is also the first appropriately powered study of varenicline not sponsored by the manufacturer, Pfizer. Subsequently, these results provide a real-world evaluation of varenicline for the inpatient setting, particularly given that patients randomized to the varenicline plus counselling arm were required to pay for the study drug themselves as this was not supplied as part of the trial.

Long-term efficacy of smoking cessation pharmacotherapy has been debated, with very few appropriately powered studies examining prolonged treatment effectiveness (beyond 52 weeks). One prospective cohort study conducted in 787 recent quitters over five years found no difference in the odds of relapse between patients who used nicotine replacement therapy for more than six weeks and those without pharmacotherapeutic assistance [21]. A placebo controlled trial published in 1971 evaluating cytisine, a drug similar to varenicline in that it acts on the nicotinic acetylcholine receptor pathway, also examined treatment efficacy at 104

**Table 2. Mortality and self-reported adverse events within 12 week treatment phase.**

|  | STOP trial (inpatients with smoking related illness) | | Pfizer [19] (Outpatient 'healthy' volunteers) | |
|  | VT+C (n = 196) | C-alone (n = 196) | Varenicline (n = 5072) | Placebo (n = 3449) |
| --- | --- | --- | --- | --- |
| Mortality | 5 (2.55) | 2 (1.02) | 6 (1.7) [20] | 1 (0.3) [20] |
| Nausea | 32 (16.33) | 3 (1.53) | 1430 (28.2) | 335 (9.7) |
| Abnormal dreams | 12 (6.12) | 2 (1.02) | 543 (10.7) | 145 (4.2) |
| Headache | 12 (6.12) | 3 (1.53) | 751 (14.8) | 410 (11.9) |
| Insomnia | 10 (5.10) | 4 (2.04) | 715 (14.1) | 300 (8.7) |
| Vomiting | 8 (4.08) | 1 (0.51) | 228 (4.5) | 66 (1.9) |
| Dizziness | 4 (2.04) | 1 (0.51) | 249 (4.9) | 197 (5.7) |

Reported as n and (%); Pfizer data from outpatient health volunteers [19]; Mortality data from Pfizer using a different cohort of participants (n = 353 in varenicline arm and n = 350 in placebo arm) [20]

**Table 3. All-cause mortality during the 52 and 104 week follow-up periods.**

|  | VT+C | C-alone |
|---|---|---|
| Arrhythmic event | 1 | 0 |
| Bradycardia | 1 | 0 |
| Lung cancer | 0 | 2 |
| Non ST segment Myocardial Infarction | 2 | 4 |
| Respiratory failure/COPD | 2 | 0 |
| Stroke | 0 | 1 |
| **Total deaths ≤ 12 months** | **6** | **7** |
| Charleston: age unadjusted (mean ± SD) | 3.29 ± 2.43 | 2.43 ± 1.62 |
| Charleston: age adjusted (mean ± SD) | 5.14 + 2.61 | 3.86 ± 2.12 |
| Lung cancer | 1 | 2 |
| Respiratory failure/COPD | 1 | 2 |
| Peripheral vascular disease | 2 | 1 |
| **Total deaths between >12 and ≤ 24 months** | **4** | **5** |
| Charleston: age unadjusted (mean ± SD) | 4.25 ± 3.20 | 4.40 ± 4.28 |
| Charleston: age adjusted (mean ± SD) | 5.50 + 3.70 | 5.80 ± 4.44 |

Number of events per arm; total sample size in each arm n = 196

week follow-up [22] producing similar results to those observed in this study (odds ratio of cytisine study 1.77; 95%CI 1.29 to 2.43; odds ratio of STOP study 1.78; 95%CI 1.10 to 2.86). There is insufficient information on the various smoking cessation products available to determine prolonged treatment efficacy, though products which substitute nicotine from cigarettes (such as transdermal nicotine patches) may be less effective than those facilitating complete removal of nicotine from the body.

## Limitations of the study

Limitations relating to the lack of blinding (resulting in both a placebo and demoralization effect) and issues pertaining to generalizability (due to predominant non-Caucasian population who are highly motivated to quit), have been described previously [11]. The lack of placebo in the control arm, as discussed previously [11], further limits the reliability of efficacy at 104-week follow-up. However, inclusion of real-world intervention arms allows for true representation of how the package of VT+C compared to C-alone will perform in the acute hospital setting, which will be of more use to policy makers and clinicians. In addition, there is the economic principal of loss aversion [23], that could be influencing the superior quit attempts observed among the VT+C arm due to the financial commitment made by these participants when purchasing the quit medication. People experience loss about twice as strongly compared to benefits from a gain of equal magnitude [24], e.g., paying $5 for the medication would be equivalent to earning $10 from a successful quit attempt. Therefore, the financial commitment made by VT+C participants has likely impacted quit efficacy in favour of the VT+C group. Of note, participants were aware of the need to purchase medication if randomised to the VT+C group.

There is an additional limitation of self-reporting treatment efficacy at 104 weeks follow-up that may overestimate the efficacy in favour of VT+C. Self-reporting of cessation among both cardiac patients [25] and those with chronic obstructive pulmonary disease [26] have suggested underreporting of true smoking status. Therefore, results need to be interpreted with caution. Of note the sample size was calculated based on the primary outcome of 52 weeks

follow-up, rather than 104 weeks. Low statistical power reduces the likelihood that statistically significant results reflect the true effect of the intervention leading to potential overestimate of the effect size and low reproducibility of results [27]. However, given the large effect size observed at 104 week follow-up the reliability of findings can be considered true and accurate.

In conclusion, the STOP trial has provided a real-world evaluation of varenicline tartrate plus counselling compared to counselling alone for administration within the inpatient setting. It has demonstrated clinically and statistically significant prolonged smoking abstinence that is well tolerated by inpatients with no increased risk of adverse events despite presentation with an acute illness episode. We suggest varenicline tartrate plus counselling be considered for standard practice among hospitalized smokers.

## Acknowledgments

We would like to acknowledge the staff of the Queen Elizabeth Hospital, Lyell McEwin Hospital and Royal Adelaide Hospital that assisted in participant recruitment. We would also like to thank Pam Gluyas and Karen Boath for their assistance with participant follow-up and Rosanna McCawley and Michelle Ashley for administrative assistance.

## Author Contributions

**Conceptualization:** Kristin V. Carson-Chahhoud, Brian J. Smith, Antony J. Veale, Adrian J. Esterman.

**Data curation:** Kristin V. Carson-Chahhoud, Malcolm P. Brinn, Faisal Ameer, Kuljit Singh, Khin Hnin.

**Formal analysis:** Kristin V. Carson-Chahhoud, Malcolm P. Brinn, Adrian J. Esterman.

**Funding acquisition:** Kristin V. Carson-Chahhoud, Brian J. Smith, Malcolm P. Brinn.

**Investigation:** Kristin V. Carson-Chahhoud, Brian J. Smith, Matthew J. Peters, Malcolm P. Brinn, Kuljit Singh, Robert Fitridge, Simon A. Koblar, Jim Jannes, Antony J. Veale, Sharon Goldsworthy, Khin Hnin, Adrian J. Esterman.

**Methodology:** Kristin V. Carson-Chahhoud, Brian J. Smith, Malcolm P. Brinn, Adrian J. Esterman.

**Project administration:** Kristin V. Carson-Chahhoud, Malcolm P. Brinn.

**Resources:** Kristin V. Carson-Chahhoud, Brian J. Smith, Malcolm P. Brinn.

**Software:** Kristin V. Carson-Chahhoud, Malcolm P. Brinn.

**Supervision:** Kristin V. Carson-Chahhoud, Brian J. Smith, Matthew J. Peters, Malcolm P. Brinn, Robert Fitridge, Simon A. Koblar, Jim Jannes, Antony J. Veale, Adrian J. Esterman.

**Validation:** Kristin V. Carson-Chahhoud, Malcolm P. Brinn.

**Visualization:** Kristin V. Carson-Chahhoud, Malcolm P. Brinn.

**Writing – original draft:** Kristin V. Carson-Chahhoud.

**Writing – review & editing:** Kristin V. Carson-Chahhoud, Brian J. Smith, Matthew J. Peters, Malcolm P. Brinn, Faisal Ameer, Kuljit Singh, Robert Fitridge, Simon A. Koblar, Jim Jannes, Antony J. Veale, Sharon Goldsworthy, Khin Hnin, Adrian J. Esterman.

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
