## [Decision Letter · Decision Letter 0]

28 Nov 2019

PONE-D-19-25086

Two year efficacy of varenicline tartrate and counselling for inpatient smoking cessation (STOP study): A Randomized Controlled Clinical Trial

PLOS ONE

Dear Dr Carson-Chahhoud ,

Thank you for submitting your manuscript to PLOS ONE. After careful consideration, we feel that it has merit but does not fully meet PLOS ONE’s publication criteria as it currently stands. Therefore, we invite you to submit a revised version of the manuscript that addresses the points raised during the review process and you will find below in the Comments to the author section.

We would appreciate receiving your revised manuscript by december 30th. To enhance the reproducibility of your results, we recommend that if applicable you deposit your laboratory protocols in protocols.io, where a protocol can be assigned its own identifier (DOI) such that it can be cited independently in the future. For instructions see: http://journals.plos.org/plosone/s/submission-guidelines#loc-laboratory-protocols

We look forward to receiving your revised manuscript.

Kind regards,

Christophe Leroyer

Academic Editor

PLOS ONE

Journal Requirements:

2.  We noticed you have some minor occurrence(s) of overlapping text with the following previous publication(s), which needs to be addressed:

In your revision ensure you cite all your sources (including your own works), and quote or rephrase any duplicated text outside the Methods section. Further consideration is dependent on these concerns being addressed.

'The Department of Respiratory Medicine, The Queen Elizabeth Hospital, Adelaide,

Australia'

'The authors received no specific funding for this work.'

Additional Editor Comments (if provided):

**Comments to the Author**

1. Is the manuscript technically sound, and do the data support the conclusions?

Reviewer #1: Yes

Reviewer #2: Yes

2. Has the statistical analysis been performed appropriately and rigorously? 

Reviewer #1: Yes

Reviewer #2: Yes

3. Have the authors made all data underlying the findings in their manuscript fully available?

Reviewer #1: Yes

Reviewer #2: Yes

4. Is the manuscript presented in an intelligible fashion and written in standard English?

Reviewer #1: Yes

Reviewer #2: Yes

5. Review Comments to the Author

Reviewer #1: The document sent by the authors is of good quality apart from some typing errors to correct, including in the summary. The methodology is clearly described and the limits are well covered in the appropriate chapter. This article is original and meets the ethical criteria of the journal.My main criticism is that varenicline placebo has not been placed in the arm with only the advice to help with weaning by phone.The second reservation, identified by the authors, is that weaning is mainly declarative. The expired CO measurement could have been done at each stage of the study and in all participants.This independent study of the laboratory that markets Varenicline shows the benefit of this molecule in long-term withdrawal and its side effects to a lesser extent than in previous studies. As such, it seems to me that this study can be published.

Reviewer #2: Comments

This study proposes to evaluate the long-term efficacy (primary objective at 2 years) and the tolerance (secondary objective) of the addition of varenicline to management by advice without medication (advice and telephone support without medication)

This is a long-term evaluation of a previous study whose effects were published in 2013 and 2014 (ref 10 and 11) after 12 months of treatment.

Usually, varenicline appears to be even more effective than nicotine substitutes for smoking cessation assistance. Alerts concerning possible neuropsychological and vascular undesirable effects, subsequently invalidated, have led to underutilization and mistrust on the part of some health professionals with regard to this molecule.

This study, which suggests a long-term efficacy of varenicline and an acceptable tolerance without excess mortality in patients with tobacco-related diseases delivers an interesting message

Another highlight of this study concerns its evaluation in patients hospitalized for a pathology secondary to smoking.

Initial inclusion: a large number of patients were excluded or not included at the beginning of the study. It would be interesting to know the main causes of non-inclusion, especially for neuropsychic diseases. This element is important since these side effects under varenicline were the subject of a debate. The good tolerance in this area under varenicline in the study may be related in part to a strict selection of patients with neuropsychological antecedents.

As noted by the authors, the strength of this study was not identified for a 2-year efficacy demonstration and a large number of patients are not assessable by that date.

Another limiting factor of the study concerns the absence of placebo versus varenicline (non-blind study) which may be negatively perceived by patients included in the group without varenicline and for health professionals. This limit is for efficacy compared to 1 and 2 years, but less alters tolerance and mortality data at 2 years (which is one of the important data in this publication).

Was there specific monitoring of varenicline patients during the first 12 weeks of follow-up?

A number of patients in the varenicline-free group eventually received varenicline and / or nicotine replacement therapy, probably during follow-up. Was this element contraindicated a priori in the context of the study?

Was treatment with varenicline systematically interrupted at 12 weeks or were some patients able to continue treatment for longer?

Deaths reported in table 2 to 12 weeks appear numerically larger under varenicline (5 versus 2). Is this element significant?

How many patients did they have a CO test to validate their self-reporting?

Patients in the varenicline group paid for their medication. This treatment is reimbursed in other countries. Can this element be an additional motivation factor for motivation and therefore effectiveness in this group?

6. PLOS authors have the option to publish the peer review history of their article (what does this mean?). If published, this will include your full peer review and any attached files.

Reviewer #1: Yes: JD. Dewitte

Reviewer #2: No

---

## [Author Response · Author response to Decision Letter 0]

1 Jan 2020

Response to editor and reviewer comments have been provided in the attachment titled 'response to reviewers' using a table format. We have provided a response to each point below as well. 

The manuscript has been reformatted to meet the PLOS ONE’s style requirements, including those for file naming. 

2. We noticed you have some minor occurrence(s) of overlapping text with the following previous publication(s), which needs to be addressed: In your revision ensure you cite all your sources (including your own works), and quote or rephrase any duplicated text outside the Methods section. Further consideration is dependent on these concerns being addressed. 

We put the manuscript through the plagiarism detection software (Turnitin), identifying some overlap with our previous 12-month efficacy publication and our published abstracts submitted to scientific conferences. Where appropriate, we have referenced the 12-month efficacy manuscript and rephrased duplicated text outside of the methods section. In particular, we have revised the introduction opening paragraph to reduce overlap with our previous 12-month efficacy publication, added the reference to our publication for methodology, and referenced table 1 for baseline demographics. For table 2 and table 3, we have not referenced this, as there is overlapping text related to some of the 12-week/baseline variables (not all of them as we are also presenting new variables) with the comparisons at 2-year follow-up all being completely new to this manuscript. We have not referenced our own abstracts, which is where much of the duplication outside of our safety (Carson 2014 et.al.) and 12-month efficacy (Smith 2012) references are coming from. 

3. Thank you for stating the following in the Acknowledgments Section of your manuscript: 'The Department of Respiratory Medicine, The Queen Elizabeth Hospital, Adelaide, Australia' We note that you have provided funding information that is not currently declared in your Funding Statement. However, funding information should not appear in the Acknowledgments section or other areas of your manuscript. We will only publish funding information present in the Funding Statement section of the online submission form. Please remove any funding-related text from the manuscript and let us know how you would like to update your Funding Statement. Currently, your Funding Statement reads as follows: 'The authors received no specific funding for this work.'

We have removed funding from the manuscript. The statement of ‘no specific funding’ would be correct for the publication, as we never received any specific funding, rather the department of respiratory medicine supported the research through personnel time as part of usual practice. 

Reviewer 1: typing errors to correct, including in the summary 

Amended

My main criticism is that varenicline placebo has not been placed in the arm with only the advice to help with weaning by phone 

We have previously mentioned the lack of placebo as a limitation of this study design, and briefly touched on it again in the limitations on page 13. To reiterate this point, we have more information about the lack of placebo in the limitations section, highlighting the reason for this specific design is to provide a real-world study for use by policy makers and clinicians. 

The second reservation, identified by the authors, is that weaning is mainly declarative. The expired CO measurement could have been done at each stage of the study and in all participants. This independent study of the laboratory that markets Varenicline shows the benefit of this molecule in long-term withdrawal and its side effects to a lesser extent than in previous studies.

Agree, this is an important limitation of this study, which has been highlighted in the discussion under limitation. We have not made any further changes at this point, however, happy to amend further if required. 

Reviewer 2: Initial inclusion: a large number of patients were excluded or not included at the beginning of the study. It would be interesting to know the main causes of non-inclusion, especially for neuropsychic diseases. This element is important since these side effects under varenicline were the subject of a debate. The good tolerance in this area under varenicline in the study may be related in part to a strict selection of patients with neuropsychological antecedents. 

Unfortunately, we were not given ethics approval to capture that level of detailed information from participants. 

Another limiting factor of the study concerns the absence of placebo versus varenicline (non-blind study) which may be negatively perceived by patients included in the group without varenicline and for health professionals. This limit is for efficacy compared to 1 and 2 years, but less alters tolerance and mortality data at 2 years (which is one of the important data in this publication). 

We have previously mentioned the lack of placebo as a limitation of this study design, and briefly touched on it again in the limitations on page 13. To reiterate this point, we have more information about the lack of placebo in the limitations section, highlighting the reason for this specific design is to provide a real-world study for use by policy makers and clinicians.

Was there specific monitoring of varenicline patients during the first 12 weeks of follow-up?

Yes, monitoring occurred at day 3, day 5, week 1, 2, 3, 4 and 12. This has been reported in the 12-month efficacy manuscript by Smith et.al. 2012.

A number of patients in the varenicline-free group eventually received varenicline and / or nicotine replacement therapy, probably during follow-up. Was this element contraindicated a priori in the context of the study? 

Yes, it was contraindicated. Participants were asked to only use the treatment assigned to them through randomisation. However, deviations from the protocol did occur and were monitored. This has been reported in the safety manuscript by Carson et.al. 2014.

Was treatment with varenicline systematically interrupted at 12 weeks or were some patients able to continue treatment for longer?

For this study, all participants reported discontinuation of treatment after 12-weeks. 

Deaths reported in table 2 to 12 weeks appear numerically larger under varenicline (5 versus 2). Is this element significant?

No, it is not significant. ‘not statistically significant’ has been added to the text of the manuscript on page 10.

How many patients did they have a CO test to validate their self-reporting?

CO testing occurred in 28 VT+C participants and 23 C-alone participants. This has been reported in the Carson et.al. 2014 publication. 

Patients in the varenicline group paid for their medication. This treatment is reimbursed in other countries. Can this element be an additional motivation factor for motivation and therefore effectiveness in this group?

Yes, we agree the addition of payment could be a motivating factor. We have now mentioned this in the limitations with supporting references relating to the economic principle of loss aversion.

---

## [Decision Letter · Decision Letter 1]

28 Jan 2020

PONE-D-19-25086R1

Two year efficacy of varenicline tartrate and counselling for inpatient smoking cessation (STOP study): A Randomized Controlled Clinical Trial

PLOS ONE

Dear A/Prof Carson-Chahhoud,

Thank you for submitting your manuscript to PLOS ONE. After careful consideration, we feel that it has merit but does not fully meet PLOS ONE’s publication criteria as it currently stands. Therefore, we invite you to submit a revised version of the manuscript that addresses the points raised during the review process.

Your manuscript has now been reviewed by a statistical reviewer (in accordance with PLOS ONE policies for manuscripts relating to Clinical Trials) - the reviewer's comments can be found below. Please accept our apologies that this feedback was not provided to you in the first review round - this was due to an oversight by journal staff.

We would appreciate receiving your revised manuscript by Mar 13 2020 11:59PM. To enhance the reproducibility of your results, we recommend that if applicable you deposit your laboratory protocols in protocols.io, where a protocol can be assigned its own identifier (DOI) such that it can be cited independently in the future. For instructions see: http://journals.plos.org/plosone/s/submission-guidelines#loc-laboratory-protocols

We look forward to receiving your revised manuscript.

Kind regards,

Artur Arikainen, Associate Editor, PLOS ONE

on behalf of

Christophe Leroyer

Academic Editor

PLOS ONE

Reviewers' comments:

Reviewer's Responses to Questions

**Comments to the Author**

1. If the authors have adequately addressed your comments raised in a previous round of review and you feel that this manuscript is now acceptable for publication, you may indicate that here to bypass the “Comments to the Author” section, enter your conflict of interest statement in the “Confidential to Editor” section, and submit your "Accept" recommendation.

Reviewer #3: All comments have been addressed

2. Is the manuscript technically sound, and do the data support the conclusions?

Reviewer #3: Yes

3. Has the statistical analysis been performed appropriately and rigorously? 

Reviewer #3: No

4. Have the authors made all data underlying the findings in their manuscript fully available?

Reviewer #3: Yes

5. Is the manuscript presented in an intelligible fashion and written in standard English?

Reviewer #3: Yes

6. Review Comments to the Author

Reviewer #3: Major;

Methods

1) Was this a registered trial on clinical trials.gov if so, should be stated in methods section.

2) Reference of protocol should be given, not just saying published in NEJM?

3) More detail required on the randomisation process, i.e was it simple randomisation, any block sizes?

Statistical analysis

1) Sample size calculation, says clustering effect was adjusted for in the sample size, the rho needs to be reported to be able to replicate the sample size?

2) If the sample size was based on the 52 weeks, then the primary analysis should based on 52 weeks with results at 104 also presented as part of the primary outcome analysis. Otherwise this needs to be mentioned if the hypothesis is that the treatment effect would be maintained at 2 years?

3) Needs more information about how the dependant variable is defined.

4) Analysis population, needs to be clearly defined, i.e. ITT population?

5) It looks like logistic regression analysis was carried out, but no mention of this in the statistical analysis section. Also based some reported results it looks like repeated measures were taken in which case, perharps more appropriate analysis is suitable to take into account the nature of the data. Also if multiple tests were done and multiple timepoints the adjustments of the p-values need to be done for these.

6) More detail on what baseline adjustment were made by discipline?

Outcome and data collection;

1) The authors state the primary outcome was continuous outcome between weeks 2 and week 104 defined smoking at ≤ 5 cigarettes in total during the follow-up periods – was this a one-off questionnaire or question asked during the phone call to participants? Also from looking at Figure 2, this implies primary outcome was collected repeatedly at various time-points? And starting from 4 weeks instead of the mentioned from 2 weeks, please clarify.

Minor;

1) Abstract in results section: “odds ratio odds ratio” repeated twice.

Introduction

1) At the end, it would be worthwhile mentioning/distinguishing primary and secondary objectives.

Results

1) It would be good add the range for age.

2) If adjustments were baseline characteristics within disciplines, then it would be helpful to present (as supplementary) a baseline table stratified by discipline to the factors adjusted for are transparent.

Outcome and data collection;

2) The sentence relating to participants assigned to intervention paying the full PBS subsided costs- does this mean they had to pay for their medication, would this have influenced/biased the primary outcome, in that motivation to stop smoking because they spent money themselves? I know it is acknowledged as a limitation, were the participants asked or aware during the consent process?

7. PLOS authors have the option to publish the peer review history of their article (what does this mean?). If published, this will include your full peer review and any attached files.

Reviewer #3: No

---

## [Author Response · Author response to Decision Letter 1]

6 Mar 2020

Major

Methods:

1) Was this a registered trial on clinical trials.gov if so, should be stated in methods section. Yes, it was registered with clinicaltrial.gov; Now added to methods. 

2) Reference of protocol should be given, not just saying published in NEJM? Amended protocol to be the clinicaltrial.gov protocol, as this is the protocol that was being referred to (though in retrospect very ambiguously).

3) More detail required on the randomisation process, i.e was it simple randomisation, any block sizes? Simple randomisation with permuted blocks of 20, now amended in the manuscript

Statistical analysis

1) Sample size calculation, says clustering effect was adjusted for in the sample size, the rho needs to be reported to be able to replicate the sample size? Rho of 0.02 as per Campbell 2000 used; Now included with reference added. 

2) If the sample size was based on the 52 weeks, then the primary analysis should based on 52 weeks with results at 104 also presented as part of the primary outcome analysis. Otherwise this needs to be mentioned if the hypothesis is that the treatment effect would be maintained at 2 years? The sample size calculation was only based on 52 week follow-up. This was published separately. The sample size was not powered to evaluate efficacy at 104-week follow-up. This has been reported in the manuscript. 

3) Needs more information about how the dependant variable is defined. Now provided for treatment efficacy as the dependant variable. 

4) Analysis population, needs to be clearly defined, i.e. ITT population? This is already reported on page 8 as being ‘intention to treat’. 

5) It looks like logistic regression analysis was carried out, but no mention of this in the statistical analysis section. Also based some reported results it looks like repeated measures were taken in which case, perharps more appropriate analysis is suitable to take into account the nature of the data. Also if multiple tests were done and multiple timepoints the adjustments of the p-values need to be done for these. There were no logistic regression or repeated measure analyses undertaken. 

6) More detail on what baseline adjustment were made by discipline? Amended in text to specify adjustment was for imbalance in people assigned to VT+C arm of the study from the vascular discipline. 

Outcome and data collection

1) The authors state the primary outcome was continuous outcome between weeks 2 and week 104 defined smoking at ≤ 5 cigarettes in total during the follow-up periods – was this a one-off questionnaire or question asked during the phone call to participants? Also from looking at Figure 2, this implies primary outcome was collected repeatedly at various time-points? And starting from 4 weeks instead of the mentioned from 2 weeks, please clarify. A one-off questionnaire at 104-weeks determined the outcome of continuous abstinence for this manuscript. Re-worded in text to be clearer. 

The outcome measure of 104-week continuous abstinence was not repeated at each time point, but only at 104-weeks. 

Minor

Abstract:

1) Abstract in results section: “odds ratio odds ratio” repeated twice. Amended

Introduction:

1) At the end, it would be worthwhile mentioning/distinguishing primary and secondary objectives. This has been amended to better reflect that 104 week follow-up was a secondary objective for the study, but evaluation of this outcome was the result presented in this manuscript. 

Results:

1) It would be good add the range for age. Added to table 1

2) If adjustments were baseline characteristics within disciplines, then it would be helpful to present (as supplementary) a baseline table stratified by discipline to the factors adjusted for are transparent. A break-down by disciplines are already presented at the end of table 1. At this stage we don’t feel that an additional breakdown of demographic characteristics will add anything further across each of the four disciplines, above what is presented in table 1.

Outcome and data collection:

2) [sic.] The sentence relating to participants assigned to intervention paying the full PBS subsided costs- does this mean they had to pay for their medication, would this have influenced/biased the primary outcome, in that motivation to stop smoking because they spent money themselves? I know it is acknowledged as a limitation, were the participants asked or aware during the consent process? Yes, participants did pay for their medication, which we do acknowledge in the limitations as a potential factor influencing their quit attempts. They were aware during the consent procedure and this has been added as a sentence to study limitations.

---

## [Editor Report · Decision Letter 2]

17 Mar 2020

Two-year efficacy of varenicline tartrate and counselling for inpatient smoking cessation (STOP study): A Randomized Controlled Clinical Trial

PONE-D-19-25086R2

Dear Dr. % Carson-Chahhoud,

We are pleased to inform you that your manuscript has been judged scientifically suitable for publication and will be formally accepted for publication once it complies with all outstanding technical requirements.

With kind regards,

Christophe Leroyer

Academic Editor

PLOS ONE

---

## [Editor Report · Acceptance letter]

16 Apr 2020

PONE-D-19-25086R2 

Two-year efficacy of varenicline tartrate and counselling for inpatient smoking cessation (STOP study): A randomized controlled clinical trial 

Dear Dr. Carson-Chahhoud:

I am pleased to inform you that your manuscript has been deemed suitable for publication in PLOS ONE. Congratulations! Your manuscript is now with our production department. 

With kind regards,

on behalf of

Dr. Christophe Leroyer 

Academic Editor

PLOS ONE